# Structural and thermodynamic insights into antibody light chain tetramer formation through 3D domain swapping

Takahiro Sakai[1], Tsuyoshi Mashima[1], Naoya Kobayashi[1], Hideaki Ogata [2], Lian Duan[3,4], Ryo Fujiki[3], Kowit Hengphasatporn[3], Taizo Uda[5], Yasuteru Shigeta [3], Emi Hifumi [6] & Shun Hirota [1] ✉

Overexpression of antibody light chains in small plasma cell clones can lead to misfolding and aggregation. On the other hand, the formation of amyloid fibrils from antibody light chains is related to amyloidosis. Although aggregation of antibody light chain is an important issue, atomic-level structural examinations of antibody light chain aggregates are sparse. In this study, we present an antibody light chain that maintains an equilibrium between its monomeric and tetrameric states. According to data from X-ray crystallography, thermodynamic and kinetic measurements, as well as theoretical studies, this antibody light chain engages in 3D domain swapping within its variable region. Here, a pair of domain-swapped dimers creates a tetramer through hydrophobic interactions, facilitating the revelation of the domain-swapped structure. The negative cotton effect linked to the β-sheet structure, observed around 215 nm in the circular dichroism (CD) spectrum of the tetrameric variable region, is more pronounced than that of the monomer. This suggests that the monomer contains less β-sheet structures and exhibits greater flexibility than the tetramer in solution. These findings not only clarify the domain-swapped structure of the antibody light chain but also contribute to controlling antibody quality and advancing the development of future molecular recognition agents and drugs.

In recent times, monoclonal antibodies have demonstrated notable success in both pharmaceutical and medical applications[1-4]. However, these antibodies often form aggregates, compromising their antigen-recognition capabilities[5,6]. Such aggregates may result from partial denaturation and nonspecific interactions of hydrophobic residues exposed to the solvent, which can serve as nuclei for larger aggregates[7,8]. Protein misfolding and aggregation can lead to various diseases[9], including amyloidosis, with immunoglobulin light-chain amyloidosis being one of the most prevalent. Overexpressed antibody light chains from small plasma cell clones can misfold and aggregate, leading to the deposition of these aggregates in tissues as amyloid fibrils, thus causing amyloidosis[10,11]. Overexpressed light chains, also known as Bence-Jones proteins, can associate with each other to form homodimers[12,13].

[1]Division of Materials Science, Graduate School of Science and Technology, Nara Institute of Science and Technology, 8916-5 Takayama, Ikoma, Nara 630-0192, Japan. [2]Graduate School of Science, University of Hyogo, 3-2-1 Koto, Kamigori-cho, Ako-gun, Hyogo 678-1297, Japan. [3]Center for Computational Sciences, University of Tsukuba, 1-1-1 Tennodai, Tsukuba, Ibaraki 305–8577, Japan. [4]Graduate School of Pure and Applied Sciences, University of Tsukuba, 1-1-1 Tennodai, Tsukuba, Ibaraki 305–8571, Japan. [5]Nanotechnology Laboratory, Institute of Systems, Information Technologies and Nanotechnologies (ISIT), 4-1 Kyudai-Shinmachi, Fukuoka 879-5593, Japan. [6]Institute for Research Management, Oita University, 700 Dannoharu, Oita-shi, Oita 870-1192, Japan. ✉e-mail: hirota@ms.naist.jp

3D domain swapping (3D-DS) is a process where the same structural region is exchanged three-dimensionally between molecules of the same protein[14,15]. The structural unit of the 3D-DS oligomers mirrors the corresponding monomer structure, except for the hinge region that connects the exchanged regions. This phenomenon is common in proteins[14–19], including amyloidogenic proteins[20]. Protein dimerization by 3D-DS tends to accelerate aggregation into amyloid-like fibrils[21–23]. Several 3D-DS structures have also been reported for certain antibodies. For instance, the human antibody 2G12, an HIV-1 neutralizer, can undergo 3D-DS by exchanging the heavy chain variable region between two antigen-binding (Fab) regions[24]. 3D-DS of exchanging the light chain variable region between two Fab regions has also been suggested[25]. The single domain VH from camelid heavy chain (VHH) can dimerize by engaging in 3D-DS for the C-terminal β-strand, where the hinge loop region converts to a β-strand in the dimer[26]. Despite extensive studies on antibody aggregation, atomic-level detail on this phenomenon remains limited.

Human kappa light chains may exhibit the structural characteristics essential for high-affinity antigen binding[27]. Hifumi et al. developed kappa light chains with catalytic activity[28,29], where light chain #4

demonstrated various conformations and aggregates[30]. Here, we illustrate the atomic-level 3D-DS structure of an antibody light chain's variable region. To explore its oligomerization properties, we utilize antibody light chain #4C214A (amino acids numbered via the Kabat method[31] Fig. 1a), in which the Cys residue forming a disulfide bond with a heavy chain Cys residue is replaced with Ala. This avoids mixing of 3D-DS dimers and disulfide-bonded dimers, allowing accurate analysis and interpretation. We also remove the constant region from #4C214A to obtain the variable region (#4V_L), which forms a 3D-DS dimer. A pair of 3D-DS dimers interacts further, forming a tetramer through hydrophobic interactions.

## Results and discussion
### Size exclusion chromatography analysis of antibody light chain #4C214A
In the size exclusion chromatography (SEC) chromatogram of #4C214A, we observed two peaks positioned around 13.1 and 16.4 mL, indicative of two distinct molecular sizes (Fig. 1b). As the protein concentration was increased, the peak area ratio of the higher-molecular-mass species to the lower molecular-mass species also

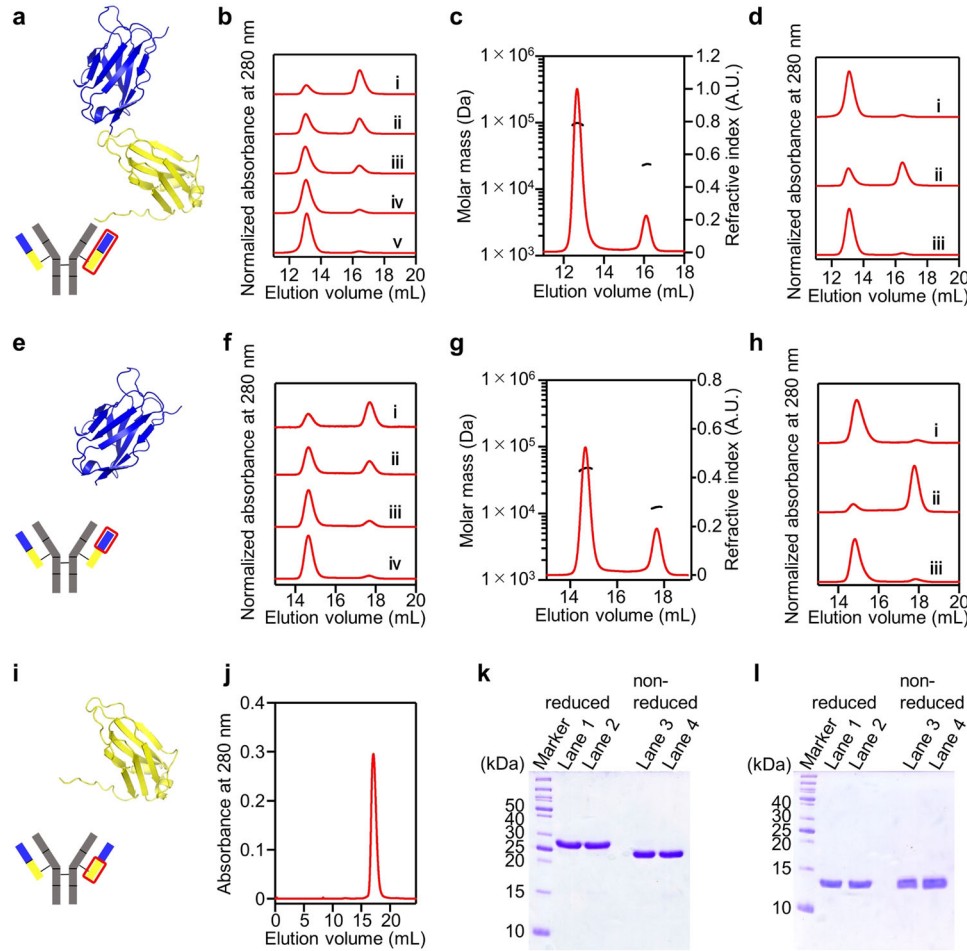

**Fig. 1 | Analyses of antibody light chain #4C214A, variable region #4V_L, and constant region #4C_L using SEC, SEC-MALS, and SDS-PAGE. a** Schematic diagram of #4C214A (top) and an antibody (bottom). **b** SEC elution profiles of #4C214A at different concentrations: (i) 3 μM, (ii) 10 μM, (iii) 30 μM, (iv) 100 μM, (v) 300 μM. **c** SEC-MALS analysis of #4C214A, displaying the molar mass (black) and refractive index (red). **d** SEC elution profiles of #4C214A at different concentrations: (i) 300 μM, (ii) post-dilution (6 μM), (iii) post-reconcentration (300 μM). **e** Schematic diagram of #4V_L (top) and an antibody (bottom). **f** SEC elution profiles of #4V_L at different concentrations: (i) 10 μM, (ii) 30 μM, (iii) 100 μM, (iv) 300 μM. **g** SEC-MALS analysis of #4V_L, displaying the molar mass (black) and refractive index (red). **h** SEC

elution profiles of #4V_L under the same conditions as in Fig. 1d. **i** Schematic diagram of #4C_L (top) and an antibody (bottom). **j** SEC elution profile of #4C_L at 300 μM. **k** SDS-PAGE gel image of #4C214A. Lane 1: higher-molecular-mass species post-treatment with 2-mercaptoethanol; Lane 2: lower-molecular-mass species post-treatment with 2-mercaptoethanol; Lane 3: higher-molecular-mass species without 2-mercaptoethanol treatment; Lane 4: lower-molecular-mass species without 2-mercaptoethanol treatment. **l** SDS-PAGE gel image of #4V_L with conditions identical to those in Fig. 1k. Cartoon model structures of #4C214A, #4V_L, and #4C_L were generated using AlphaFold2. Source data are provided as a Source Data file. Each analysis has been performed at least 3 times.

increased (Fig. 1b). Subsequent SEC-multi angle light scattering (MALS) analysis revealed the molecular masses of the smaller and larger SEC elution volume samples to be approximately 97.1 and 23.6 kDa, respectively (Fig. 1c). These values were in good agreement to the molecular masses of the tetramer (101 kDa) and monomer (25.2 kDa), respectively. However, in the SDS-PAGE analysis, single bands were detected at approximately 27 and 25 kDa in the presence and absence of 2-mercaptoethanol, respectively, for both the higher- and lower-molecular-mass species (Fig. 1k). This observation suggests that the higher-molecular-mass species is not formed by an intermolecular disulfide bond but is likely a non-covalent tetramer.

The ratio of monomer to tetramer was 4:96 in a 300 μM solution of #4C214A after incubation at 25 °C for 24 h (Fig. 1d, curve i). However, this ratio shifted to 40:60 when the solution was diluted to 6 μM of #4C214A and incubated under the same conditions (Fig. 1d, curve ii). Upon reconcentration of the diluted #4C214A solution to 300 μM and further incubation at 25 °C for 24 h, the ratio reverted back to 4:96 (Fig. 1d, curve iii). These results indicate that #4C214A can reversibly associate and dissociate between the monomeric and tetrameric states, depending on the protein concentration. Significantly, only monomers and tetramers were observed, with no detection of dimers, trimers, or higher-order oligomers, suggesting that #4C214A oligomerizes through specific interactions.

### Preparation and oligomerization assessment of #4V$_L$ and #4C$_L$

To determine whether the variable or constant region contributed to the oligomerization of #4C214A, we cleaved the antibody light chain and individually assessed the oligomerization properties of the variable and constant regions. We inserted a tobacco etch virus (TEV) protease recognition sequence between these regions of #4C214A to create #4C214AinsCS$^{TEV}$. After purification, #4C214AinsCS$^{TEV}$ was cleaved with TEV protease to yield the free variable region (#4V$_L$) and the constant region (#4C$_L$), as shown in Supplementary Fig. 1.

Two peaks around 14.6 and 17.7 mL were observed in the SEC chromatogram of #4V$_L$ (Fig. 1f). The ratio of higher-molecular-mass to lower-molecular-mass species in #4V$_L$ increased with increase in the protein concentration, similar to the pattern observed for #4C214A (Fig. 1b, f). As per the SEC-MALS analysis, the two peaks corresponded to molecular masses of 47.1 and 12.4 kDa, respectively, a similarity shared with the #4C214A elution curve (Fig. 1c, g). Moreover, the #4V$_L$ samples that eluted at different SEC volumes displayed bands at the monomer positions in SDS-PAGE analysis, both with and without 2-mercaptoethanol (Fig. 1l). Conversely, only one peak was detected around 17.1 mL in the SEC chromatogram of #4C$_L$, indicating that #4C$_L$ molecules were not associating with each other (Fig. 1j). These results suggest that #4V$_L$ maintains an equilibrium between the monomeric and tetrameric states, which causes tetramerization of #4C214A.

### X-ray crystal structure of #4V$_L$ tetramer

The X-ray crystal structure of the #4V$_L$ tetramer has been resolved at a 2.0 Å resolution (PDB code: 8KAD). In the asymmetric unit of the #4V$_L$ crystal, two independent #4V$_L$ molecules form a single 3D-DS dimer. A pair of these 3D-DS dimers constitutes a tetramer (Fig. 2a). Within the 3D-DS dimer, the β-strand located in the C-terminal region (Leu96–Lys107) is exchanged between the protomers (Fig. 2a). When comparing the crystal structure of the 3D-DS dimer protomer to the #4V$_L$ model structure, produced using AlphaFold2[32], the hinge region is identified as residues Arg91–Pro95, which is a part of the complementarity-determining region 3 (CDR3) (Fig. 2b). Interestingly, the N-terminal residues of CDR3 (Met89, Gln90, Arg91, Ile92, and Glu93) at the hinge region form a β-strand structure that interacts with a portion of CDR1 (Asn30, Pro31, Ser32, Phe33, and Asp34), which also forms a β-strand structure (Fig. 2c). There are limited instances reported where the hinge loop transforms into a β-strand during 3D domain swapping, including cases with VHH, RNase A, and β2-

microglobulin[26,33–35]. In previously reported 3D-DS structures of immunoglobulin fold proteins, protomers interact at the hinge region (a portion of CDR3) via hydrogen bonding and/or π-cationic interactions, which stabilize the dimer[34,35]. However, in the hinge region between the protomers of the #4V$_L$ 3D-DS dimer, no significant hydrogen bonds or π-cation interactions were observed (Fig. 2b), suggesting the presence of other factors that stabilize the 3D-DS structure.

We calculated the Cα root-mean-square deviation (RMSD) values between the crystal dimer protomer structures and the model monomer structure derived from AlphaFold2. These calculations were performed for the framework regions FR1 (Asp1–Cys23), FR2 (Trp35–His49), and FR3 (Gly51–Cys88) located before the hinge loop and for the C-terminal framework region FR4 (Phe98–Lys107) located after the hinge loop, excluding the CDR domains. The RMSD values for the two protomers were 0.68 and 0.97 Å for the structure prior to the hinge loop, and 0.47 and 0.62 Å for the structure following the hinge loop. These results suggest a high degree of structural similarity between the monomer and dimer in both regions.

Within the 3D-DS dimer, we identified 29 hydrogen bonds (<3.5 Å between heavy atoms) between the protomers (Fig. 3a). Out of these, 20 were situated on the β-sheet backbone, comparable to the model monomer structure (Fig. 3b). Alongside these backbone hydrogen bonds, two symmetric hydrogen bonds formed between the main chains of the protomers in the 3D-DS dimer—bonds absent in the model monomer structure (HB1 and HB1′, where prime denotes the symmetric hydrogen bond). Two additional symmetric hydrogen bonds were formed between the main and side chains (HB2 and HB2′), and two more formed between the side chains (HB3 and HB3′) of the swapped region and the remaining region between the protomers (Fig. 3b). However, three asymmetric hydrogen bonds were detected between the swapped region and the remaining areas (HB4, HB5, and HB6). Additionally, a novel hydrogen bond, absent in the monomer, was identified at the domain interface between the dimer's protomers (HB7) (Fig. 3a).

Clusters of aromatic residues were identified between the dimers (Fig. 3c), indicating that aromatic interactions significantly contribute to the stabilization of the tetramer structure (Fig. 2a). The cluster composed of Tyr36, Pro44, Tyr87, and Phe98 forms a small cavity. These residues have been documented to construct a comparable hydrophobic core in a light-chain variable-region dimer that does not undergo 3D-DS[12], suggesting that the two 3D-DS dimers leverage a similar hydrophobic core. Moreover, 13 hydrogen bonds were observed at the interface between the 3D-DS dimers (Fig. 3c). Eleven of these hydrogen bonds encircled the aromatic cluster (Fig. 3c), presumably augmenting hydrophobic interactions and stabilizing the tetramer structure. The 3D-DS structure makes this possible, which is likely to be important for the stabilization of the tetramer. Furthermore, Arg50$_A$(Nη2), Glu93$_A$(Oε2), Arg91$_D$(Nη2), Glu93$_D$(Oε1) (subscripts A–D denote each protomer), and a water molecule form a hydrogen bond network, which likely further stabilizes the tetramer structure (Fig. 4a).

To explore the stability of the hydrogen bonds and hydrophobic interactions, we carried out residue interaction network generator (RING) and PyContact analyses on the crucial hydrogen bonds and hydrophobic interactions found in the X-ray #4V$_L$ tetramer structure using trajectories acquired from molecular dynamics (MD) simulations (Fig. 4b, c). The majority of the key hydrogen bonds and hydrophobic interactions remained stable throughout the MD simulation, excluding intraprotomer hydrogen bonds HB8 and HB18, and the interprotomer hydrogen bond HB7, which fractured over time. The disintegration of this bond was attributed to the exposure of Asp1$_B$ in the C-terminal region of protomer B to the aqueous phase. Additionally, HB12 was rapidly disrupted, and Arg50$_A$ exhibited hydrogen bonding interactions with the nearest water molecule for almost the entirety of the MD

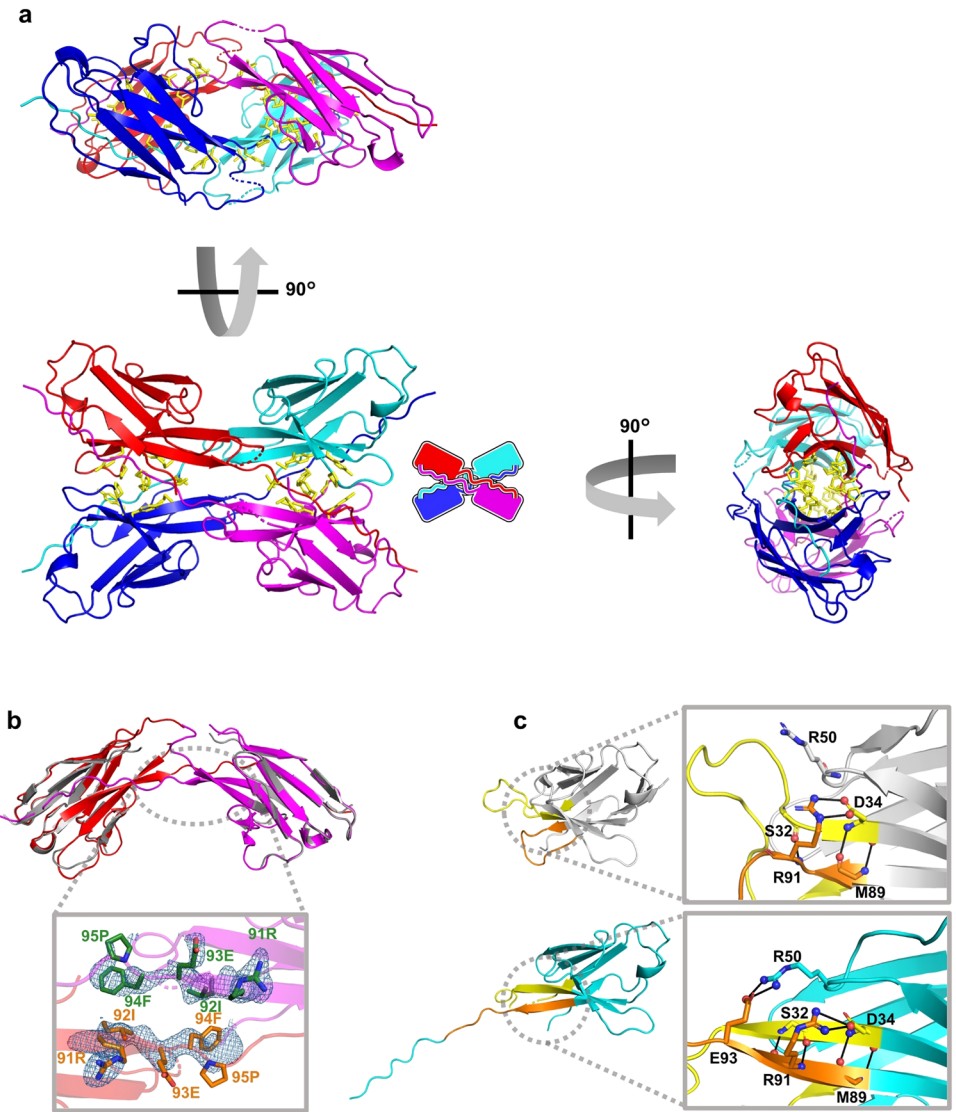

**Fig. 2 | Crystal structure of #4V$_L$. a** The crystal structure of the #4V$_L$ tetramer with its schematic representation in the right (left bottom). The right figure represents a 90-degree rotation of the left figure, while the left top figure is a 90-degree rotation of the left bottom figure. **b** An overlapped view of the #4V$_L$ dimer structure (shown in red and magenta; PDB code: 8KAD) and the #4C214A model structure produced using AlphaFold2 (gray). An enlarged view of the hinge region from below is displayed at the bottom. The side chains of residues 91 to 95 comprising the hinge region are represented as stick models, highlighted in green and orange. The nitrogen and oxygen atoms of the side chains of the hinge-loop residues are depicted in blue and red, respectively. The 2F$_o$ − F$_c$ electron density map of the hinge-loop residues is shown at contour level 1σ in blue. **c** The model monomer structure of #4V$_L$ (gray) is juxtaposed with the subunit crystal structure of the #4V$_L$ tetramer (cyan). Enlarged views of the residues, wherein the loop structure in the monomer transforms into a β-strand in the tetramer, are displayed in the right figures. Hydrogen bonds within these residues are denoted by black lines. CDR1 and CDR3 are colored in yellow and orange, respectively.

trajectory (~99%) (Fig. 4d). The hydrogen bonding interaction and water bridge involving Arg50$_A$(Nη2), Glu93$_A$(Oε2), Arg91$_D$(Nη2), Glu93$_D$(Oε1), and the water molecule enhanced stability near the CDR3 region in the tetramer structure, in which CDR3 often contains hydrophilic residues that help maintain the solubility and stability of the antibody. The breakdown of HB14 and HB16 was noted during the simulation, presumably due to their location on the intermolecular surface, rendering them vulnerable to water exposure.

Despite observing a monomer-tetramer equilibrium for #4V$_L$ (Fig. 1) and the construction of the tetramer from two 3D-DS dimers (Fig. 2), no dimers were detected in the SEC analysis. To investigate the lack of the free 3D-DS dimer of #4V$_L$, we performed 3D-RISM calculations for the tetramer and two 3D-DS dimers in explicit water molecules (Supplementary Fig. 2). The calculated total solvation free energies were −4710.89 kcal mol$^{-1}$ for the #4V$_L$ tetramer and −4441.16 kcal mol$^{-1}$ for two 3D-DS dimers (Fig. 4e). These findings indicate that the #4V$_L$ tetramer is more stable than the 3D-DS dimer by −269.73 kcal mol$^{-1}$, which corresponds to the free energy of dimer-to-tetramer formation. Additionally, hydrophobic interactions were pivotal in stabilizing the #4V$_L$ tetramer through side chain/side chain interactions of hydrophobic amino acids from all four chains, demonstrating remarkable stability throughout the MD simulations (Fig. 4c). Notably, both the potential energy and solvation free energy terms contributed to the structural stability of the #4V$_L$ tetramer compared to the 3D-DS dimers (Fig. 4e).

### Equilibrium and dissociation of #4C214A and #4V$_L$ tetramers

The dissociation constant ($K_D$) of #4C214A was determined by the ratio of the monomer and tetramer, yielding a comparable value of 2.8 ± 0.4 × 10$^{-16}$ M$^3$ across protein concentrations of 3–300 μM (Supplementary Table 1). The standard Gibbs free energy change (Δ$G$°) for the dissociation of the #4C214A tetramer was calculated from the

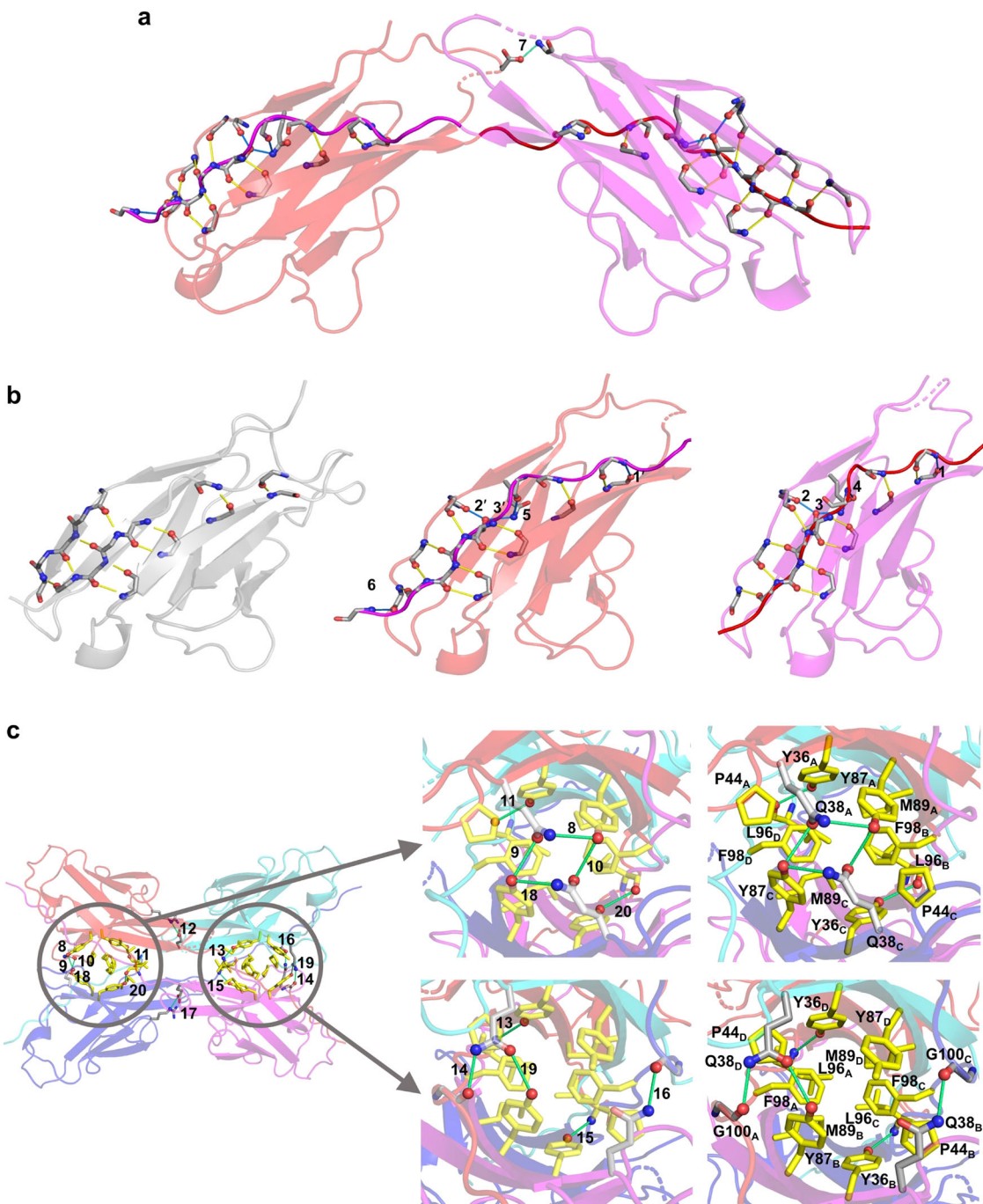

**Fig. 3 | Critical hydrogen bonds and hydrophobic residues in the #4V$_L$ tetramer. a** The 3D-DS dimer structure of #4V$_L$. The protomers of the dimer are depicted in red and magenta, with backbone hydrogen bonds represented as yellow and blue lines. The backbone hydrogen bonds in the dimer structure containing an atom in the swapping region that align with the model structure are shown in yellow, while those not observed in the model monomer structure are shown in blue. **b** Functional unit structures of the 3D-DS dimer obtained from the crystal structure (PDB code: 8KAD; each protomer is shown in red or magenta) together with the model monomer structure of #4V$_L$ (gray). The model monomer structure was produced using AlphaFold2. The hydrogen bonds are colored the same as in Fig. 3a. **c** Crystal structure of the #4V$_L$ tetramer (left). Enlarged view of the hydrophobic residues and hydrogen bonds between the 3D-DS dimers (right). Protomers A, B, C, and D are colored in red, magenta, blue, and cyan, respectively.

Hydrophobic residues are indicated as yellow sticks. Light blue lines display key hydrogen bonds. The labels HB1–HB20 represent the key hydrogen bonds (HB are omitted in the figures). HB1: Gln90$_A$(O)-Thr97$_B$(N), HB2: Thr102$_A$(Oγ1)-Pro8$_B$(O) HB3: Thr102$_A$(Oγ1)-Gln6$_B$(Nε2), HB4: Gly101$_A$(N)-Gln6$_B$(Oε1), HB5: Gln6$_A$(Oε2)-Gly100$_B$(O), HB6: Val13$_A$(O)-Lys107$_B$(N), HB7: Asp27d$_A$(Oδ2)-Asp1$_B$(N), HB8: Gln38$_A$(Nε2)-Tyr87$_A$(Oη), HB9: Gln38$_A$(Oε1)-Tyr87$_C$(Oη), HB10: Tyr87$_A$(Oη)-Gln38$_C$(Oε1), HB11: Tyr36$_A$(Oη)-Leu96$_D$(O), HB12: Arg50$_A$(Nε)-Glu93$_D$(Oε1), HB13: Leu96$_A$(N)-Tyr36$_D$(Oη), HB14: Gly100$_A$(O)-Gln38$_B$(Nε2), HB15: Tyr36$_B$(Oη)-Leu96$_C$(N), HB16: Gln38$_B$(Nε2)-Gly100$_C$(O), HB17: Glu93$_B$(Oε1)-Arg50$_C$(Nε), HB18: Gln38$_C$(Nε2)-Tyr87$_C$(Oη), HB19: Tyr87$_B$(Oη)-Gln38$_D$(Oε1), HB20: Leu96$_B$(O)-Tyr36$_C$(Oη). Key hydrogen bonds are further detailed in the figure, with symmetric hydrogen bonds labeled with a prime. Hydrogen bonds HB4–HB6 were detected only in a single unit.

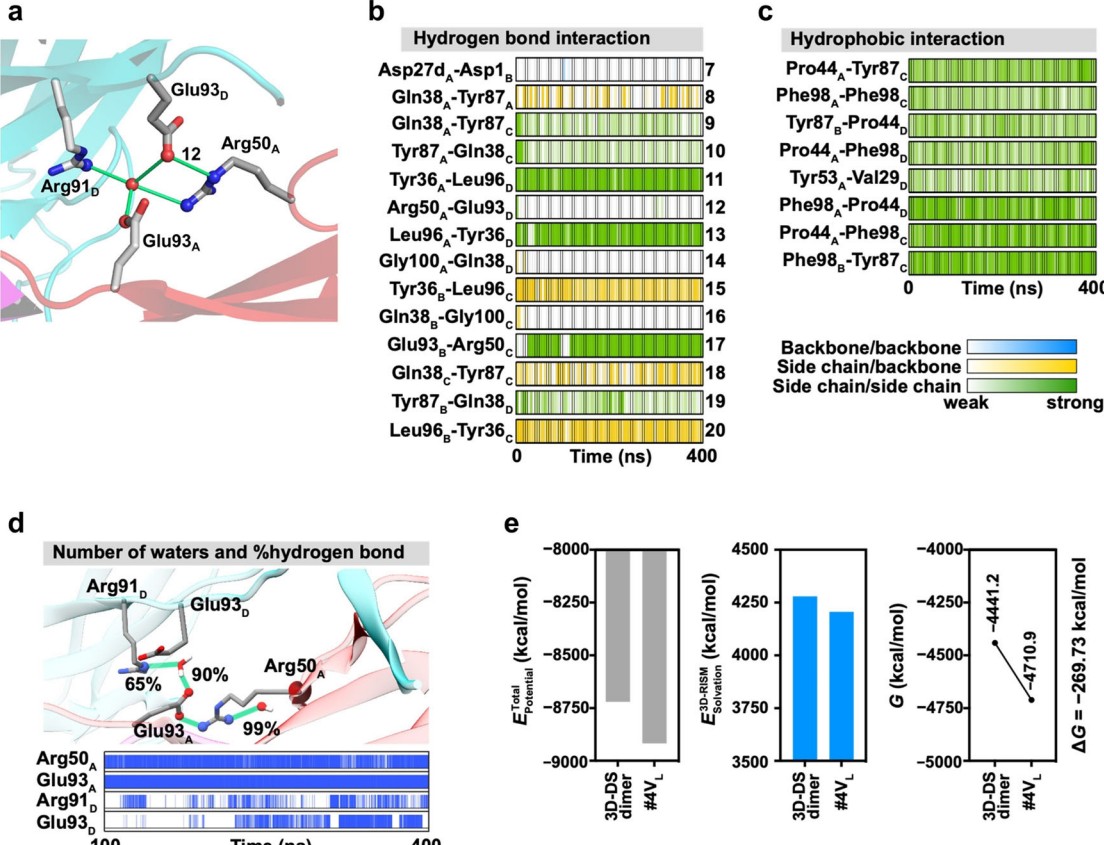

**Fig. 4 | Examination of critical hydrogen bonds and hydrophobic residues in the #4V$_L$ tetramer. a** Illustration of a hydrogen bond network formed by Arg50$_A$(Nη2), Glu93$_A$(Oε2), Arg91$_D$(Nη2), Glu93$_D$(Oε1), and a water molecule. Label "12" represents hydrogen bond HB12. **b, c** Hydrogen bonding and hydrophobic intermolecular interaction profile of #4V$_L$, as analyzed by the PyContact program. Interactions for backbone/backbone, side chain/backbone, and side chain/side chain are colored blue, yellow, and green, respectively. The strength of interactions is signified by the gradient of colors. **d** Hydrogen bond network involving Arg50$_A$(Nη2), Glu93$_A$(Oε2), Arg91$_D$(Nη2), and Glu93$_D$(Oε1) (top). The percentages represent the hydrogen bonds between the amino acid (Arg50$_A$(Nη2), Glu93$_A$(Oε2), Arg91$_D$(Nη2), or Glu93$_D$(Oε1)) and water molecules. The bar graph illustrates the

presence of water molecules in the first or second solvation shell around Arg50$_A$(Nη2), Glu93$_A$(Oε2), Arg91$_D$(Nη2), and Glu93$_D$(Oε1) (bottom). The number of water molecules was determined using distance criterion of 3.4 and 5.0 Å for the first and second solvation shells, respectively. The number of water molecules was 0 or 1 throughout all trajectories. **e** Evaluation of structural stability based on total free energy using the 3D-RISM theory. $E_{Potential}^{Total}$, $E_{Solvation}^{3D-RISM}$, and $G$ denote the total potential energy, solvation free energy, and free energy, respectively, of two 3D-DS dimers or the #4V$_L$ tetramer. $\Delta G$ signifies the free energy difference between the #4V$_L$ tetramer and two 3D-DS dimers, which equates to the tetramer formation energy from two dimers. Source data are provided as a Source Data file.

dissociation equilibrium constant $K_D$ at diverse temperatures (Supplementary Table 2). By employing Eq. (4) to fit the $\Delta G°$ versus temperature plot (Fig. 5a), we derived the standard specific heat change ($\Delta C°_p$), the temperature ($T_h$) at which the standard enthalpy change ($\Delta H°$) equals zero, and the temperature ($T_S$) at which the standard entropy change ($\Delta S°$) equals zero (Fig. 5a). These were found to be 0.78 kcal mol⁻¹K⁻¹, 1 °C, and 29 °C, respectively. These values were then substituted into Eqs. (2) and (3) to determine $\Delta H°$ and $\Delta S°$ across the temperature range of 4–45 °C (Fig. 5b). $\Delta H°$ values for #4C214A were positive throughout the temperature span and increased as temperature increased, while the $\Delta S°$ values for #4C214A transitioned from negative to positive. Generally, the $\Delta H°$ of protein oligomers bound via hydrogen bonds and/or electrostatic interactions diminishes as temperature rises[36]. These results support our hypothesis that hydrophobic interactions play a pivotal role in tetramerization (Figs. 2a and 3c). The $\Delta H°$ (2.6–34 kcal mol⁻¹) and $\Delta S°$ (−0.02–0.02 kcal K⁻¹ mol⁻¹) values of the #4C214A tetramer dissociation corresponded well to the values reported for the melittin tetramer (−5–25 kcal mol⁻¹ and −0.07–0.02 kcal K⁻¹ mol⁻¹)[37] at 15–45 °C (Fig. 5b), suggesting that the enthalpy factor is significant in the tetramerization of both #4C214A and melittin.

The $\Delta G°$ values for #4V$_L$ were calculated similarly to #4C214A from $K_D$ at temperatures ranging from 4–45 °C (Supplementary Table 2), yielding $\Delta C°_p$, $T_h$, and $T_S$ as 0.84 kcal mol⁻¹K⁻¹, 24 °C, and −1 °C, respectively (Fig. 5c). $\Delta H°$ (3.2–38 kcal mol⁻¹) and $\Delta S°$ (−0.03–0.02 kcal K⁻¹ mol⁻¹) were determined in the same manner as for #4C214A across the temperature range of 4–45 °C (Fig. 5d). These values correspond closely with those of #4C214A, further substantiating our hypothesis that #4C$_L$ does not significantly influence the tetramerization of #4C214A.

The dissociation properties of the tetramer were investigated by monitoring the tetramer-to-monomer dissociation using SEC, achieved by diluting a 300 μM solution of #4C214A fifty-fold at temperatures ranging from 4–45 °C (Fig. 5e, Supplementary Fig. 3a). The relaxation time, reflecting the time course of the tetramer ratio (Fig. 5f), was found to be between 13.5–0.3 h at 4–45 °C. This indicated that the dissociation rate constant of the #4C214A tetramer was notably slower than those of typical protein complexes, suggesting that relatively substantial structural changes are required for tetramer dissociation, a characteristic consistent with the 3D-DS structure. The relaxation time for #4V$_L$ tetramer dissociation into monomers, ascertained through SEC, was found to be 11.8–0.2 h at 4–45 °C (Fig. 5g, h,

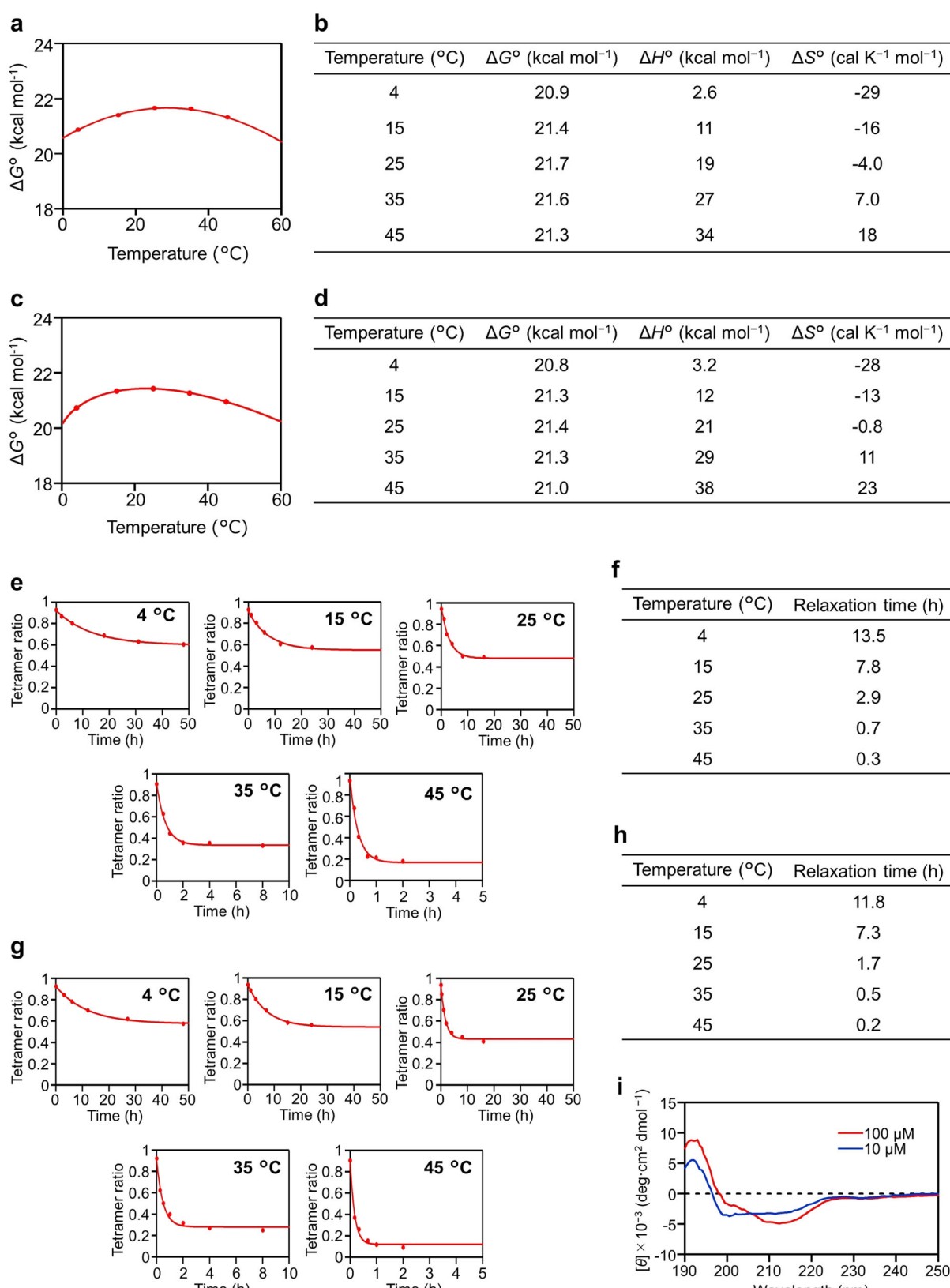

**Fig. 5 | Dissociation of #4C214A and #4V_L tetramers into monomers. a** Plot depicting the relationship between $\Delta G°$ and temperature for #4C214A, including the fitted curve based on Eq. (4). **b** Thermodynamic parameters for the monomer-tetramer equilibrium of #4C214A at 25 °C. **c** A plot similar to Fig. 5a for #4V_L. **d** A table corresponding to Fig. 5b for #4V_L. **e** Time-course tracking of the tetramer ratio change in #4C214A along with the corresponding exponentially fitted curves.

**f** Rate constants for #4C214A, illustrating the dissociation from tetramers to monomers at different temperatures. **g** Graphs similar to Fig. 5e for #4V_L. **h** A table corresponding to Fig. 5f for #4V_L. **i** CD spectra of #4V_L at different concentrations: 100 μM (red) and 10 μM (blue). Source data are provided as a Source Data file. Each analysis has been performed at least 3 times.

Supplementary Fig. 3b), a time frame similar to that of the #4C214A tetramer. This supports the hypothesis that #4C$_L$ does not significantly contribute to the tetramerization of #4C214A.

The $K_D$ of #4V$_L$ was determined to be a constant $3.0 \pm 0.2 \times 10^{-15}$ M$^3$ between 10 and 300 μM at 25 °C, as calculated using the monomer-tetramer ratio (Supplementary Table 1). This constant was slightly larger than that of #4C214A, indicating that the constant regions of #4C214A might slightly stabilize the tetramer structure through hydrophobic interactions.

The negative Cotton effect related to β-sheet formation, observed at approximately 215 nm in the circular dichroism (CD) spectrum of #4V$_L$, was more prominent at a protein concentration of 100 μM than at 10 μM (Fig. 5i). According to SEC analysis, the monomer-to-tetramer ratios of #4V$_L$ were 61:39 and 14:86 at protein concentrations of 10 and 100 μM, respectively (Supplementary Fig. 3c). These findings suggest that the disordered region of #4V$_L$ at lower protein concentrations transitions to a β-sheet structure at higher concentrations. The stabilization of #4V$_L$ secondary structures at higher concentrations appears to be driven by the monomer-to-tetramer conversion facilitated by 3D-DS. Higher secondary structure stability may prevent aggregation of the light chain that could lead to systemic amyloidosis. On the other hand, 3D-DS can occur in various manners in antibodies[24–26]. The native antibody light chain has a cysteine residue in the constant region that can causes dimerization of antibodies through a disulfide bond. The presence of various 3D-DS types together with disulfide bond linkages of two cysteine residues of different antibody light chains may promote aggregation.

In conclusion, #4C214A maintains an equilibrium between its monomeric and tetrameric states. We prepared a mutant wherein a TEV protease recognition site was incorporated between the variable and constant regions of #4C214A and cleaved the modified light chain using TEV protease, obtaining the variable region #4V$_L$ and constant region #4C$_L$. #4V$_L$ was revealed using SEC to also exist in an equilibrium between its monomeric and tetrameric states, whereas #4C$_L$ was detected only in a monomeric state. In the crystal structure, a pair of #4V$_L$ 3D-DS dimers formed a tetramer, with numerous hydrophobic residues creating a cavity at the dimer-dimer interface. The $\Delta H°$ calculated from the dissociation constants, reflecting the dissociation of the tetramer to monomers, increased with increasing temperature, thereby reinforcing the hypothesis that hydrophobic interactions play a crucial role in the tetramerization of both #4C214A and #4V$_L$. Moreover, the slow dissociation rate of the #4C214A tetramer and #4V$_L$ tetramer compared to typical protein complexes implies considerable structural changes during the dissociation process. We obtained the 3D-DS structure of an antibody light chain's variable region, resulting from the stabilization of the 3D-DS dimer through tetramer formation driven by hydrophobic interactions between two 3D-DS dimers. These results unequivocally illustrate that the light chain can undergo 3D-DS at CDR3, which might be related to prevention of antibody aggregation. Research on the 3D-DS of light chains, particularly concerning aggregation, is limited. This study lays the foundation for future research in this field.

## Methods

### Expression and purification of #4C214A

*Escherichia coli* BL21 (DE3) pLysS cells were transformed with a pET20b(+) plasmid, containing the antibody light chain gene #4C214A[38], and cultivated at 18 °C in Luria-Bertani medium supplemented with 100 mg/L ampicillin. When the OD$_{600}$ value ranged between 0.6 and 0.8, isopropyl β-D-1-thiogalactopyranoside (IPTG) was introduced into the culture at a final concentration of 10 μM. The culture was subsequently incubated at 18 °C for 16 h before being harvested by centrifugation at 8,000 g for 20 min at 4 °C. The resulting cell pellet was resuspended in a minimal volume of 25 mM Tris(hydroxymethyl)aminomethane-hydrochloric acid (Tris-HCl) buffer, pH

8.0, containing 0.25 M sodium chloride (NaCl). The *Escherichia coli* suspension was then chilled on ice and sonicated in three 2-minute intervals, with 1-second pauses, using an ultrasonic liquid processor (VC 505, SONICS&MATERIALS, Inc.). The sonicated sample was subsequently centrifuged at 18,000 g for 20 min at 4 °C. The supernatant, containing #4C214A, was purified by Ni-affinity chromatography (HisTrap HP, Cytiva) using a fast protein liquid chromatography (FPLC) system (AKTA GO, Cytiva) at a flow rate of 2.5 mL/min, with a monitoring UV-vis detector settled at 280 nm. The buffer gradient consisted of 25 mM Tris-HCl buffer, pH 8.0, containing 0.25 M NaCl and the same buffer, containing 0.25 M NaCl and an additional 0.3 M imidazole, all at 4 °C. Fractions containing #4C214A were collected and subsequently analyzed by SEC (HiLoad 26/600 Superdex 200 pg, Cytiva) using the same FPLC system at a flow rate of 1.3 mL/min. The buffer for this stage was phosphate-buffered saline (PBS) and the temperature was kept constant at 4 °C. Finally, the purified #4C214A was analyzed by conducting SDS-PAGE on a 15% acrylamide gel.

### Expression and purification of tobacco etch virus protease

The gene fragment of TEV protease was integrated into a pET-21 vector with a maltose-binding protein (MBP). The fusion protein consisted of an N-terminal MBP, succeeded by a TEV protease and a His tag. *Escherichia coli* BL21 (DE3) pLysS cells, after being transformed with the plasmid, were cultured under the same conditions previously detailed for #4C214A. The TEV protease was then isolated and harvested following the aforementioned method. The supernatant was applied to a Ni-NTA agarose column (Wako). The column was subsequently washed with 50 mM Tris-HCl buffer, pH 8.0, containing 0.25 M NaCl and 0.03 M imidazole. The TEV protease was finally eluted using 50 mM Tris-HCl buffer, pH 8.0, containing 0.25 M NaCl and 0.3 M imidazole.

### Expression and purification of variable region #4V$_L$ and constant region #4C$_L$

The TEV protease recognition motif ENLYFQG was introduced via mutagenesis between the variable and constant regions, specifically between Lys107 and Arg 108, of #4C214A (Supplementary Fig. 4) via mutagenesis, creating #4C214AinsCS$^{TEV}$ (Supplementary Fig. 5). This mutagenesis was achieved by conducting PCR on the #4C214A pET20b(+) plasmid using mutation primers (Eurofins Genomics, Japan) (Supplementary Fig. 6). *Escherichia coli* BL21 (DE3) pLysS cells were cultured and harvested using the same procedures employed for #4C214A. #4C214AinsCS$^{TEV}$ was extracted and purified following the aforementioned method.

The protease recognition sequence of purified #4C214AinsCS$^{TEV}$ was cleaved by incubation with the TEV protease (Supplementary Fig. 7) at 4 °C for 16 h in 50 mM Tris–HCl buffer, pH 8.0, which contained 0.25 M NaCl, 0.5 mM ethylenediaminetetraacetic acid (EDTA) dihydrate, and 1 mM dithiothreitol. The solution was then passed three times through the Ni-NTA agarose (Wako) column to bind #4C$_L$ (Supplementary Fig. 5) and TEV protease, which contain His tags. This process resulted in the extraction of #4V$_L$ (Supplementary Fig. 5) in the flow-through solution. Subsequently, #4C$_L$ and TEV protease were eluted with 50 mM Tris–HCl buffer, pH 8.0, containing 0.25 M NaCl and 0.3 M imidazole.

#4V$_L$ was further purified by anion exchange column chromatography performed with a HiTrap Q HP column (Cytiva) on the FPLC system (AKTA GO) with a flow rate of 2.5 mL/min, a monitoring UV-vis detector settled at 280 nm, and a gradient of 50 mM Tris–HCl buffer, pH 9.0, and the same buffer containing 1 M NaCl, at a temperature of 4 °C. #4V$_L$ was subsequently purified by SEC performed with a HiLoad 26/600 Superdex 200 pg column (Cytiva) on the same FPLC system with a flow rate of 1.3 mL/min, a monitoring UV-vis detector settled at 280 nm, PBS buffer, and at a temperature of 4 °C. Similarly, #4C$_L$ was purified using SEC under the same conditions.

The molecular masses of #4V$_L$ and #4C$_L$ were validated by liquid chromatography coupled with electrospray ionization triple quadrupole time-of-flight mass spectrometry (LC-ESI-QTOF/MS) (LCMS-9030, Shimadzu, Japan). The LC system delivered solvents at a rate of 0.2 mL/min. Gradient elution used mobile phases of 0.1% formic acid (v/v) in CH3CN (Solvent A) and 0.1% formic acid (v/v) in H$_2$O (Solvent B). The gradient was initiated at 10% B and linearly increased to 90% B over 15 min. The observed masses of #4V$_L$ and #4C$_L$ corresponded with their expected calculated values, confirming successful purification (Supplementary Fig. 1b). The purification of #4V$_L$ and #4C$_L$ was further verified with SDS-PAGE on 15% acrylamide gels (Fig. 1l and Supplementary Fig. 1a).

## SEC and SEC-multi angle light scattering measurements

The monomer-to-tetramer ratios of 300 μM #4C214A and #4V$_L$ solutions were examined by SEC with a Superdex 200 Increase 10/300 GL column (Cytiva) on a high-performance liquid chromatography (HPLC) system. The system maintained a flow rate of 0.3 mL/min, a monitoring UV-vis detector settled at 280 nm, and PBS buffer, at a temperature of 4 °C. The HPLC system was equipped with a diode array detector (SPD-M20A, Shimadzu), liquid chromatograph (LC-20AB, Shimadzu), degasser (DGU-20A3, Shimadzu), and autosampler (SIL-20AC HT, Shimadzu). The solutions were then diluted 50-fold with PBS to achieve a final concentration of 6 μM. Following a 24-h incubation at 25 °C to reach equilibrium, the solutions were re-evaluated using SEC under the same conditions. Subsequently, the solutions were concentrated again to 300 μM using an Amicon ultrafiltration tube (Merck Millipore; 10,000 NMWL for #4C214A, and 3,000 NMWL for #4V$_L$). After another 24-h incubation at 25 °C, the SEC analysis was performed again under the same conditions. The peak areas in the SEC chromatograms were obtained through least-squares fitting of the peaks with Gaussian curves, using the Igor Pro 6.37 software (WaveMetrics, Portland). The monomer-to-tetramer ratios were calculated based on the relative peak areas. The #4C214A and #4V$_L$ solutions at other concentrations (#4C214A, 3–100 μM; #4V$_L$, 10–100 μM) were incubated at 25 °C for 24 h and subsequently assessed via SEC under the same conditions.

SEC-MALS of purified #4C214A and #4V$_L$ was performed using a Superdex 200 Increase 10/300 GL column (Cytiva), a miniDawn multi-angle light scattering detector (Wyatt Technology Corporation), and a RID-20A refractive index detector (Shimadzu). The HPLC system (Shimadzu) was used at a flow rate of 0.5 mL/min with PBS buffer at a temperature of 25 °C. Each sample involved a 100 μL of 40 μM protein solution. All data processing and analysis were conducted with the Astra software (Wyatt Technologies), assuming a standard refractive index increment (dn/dc) value of 0.185 mL/g for all proteins.

## X-ray crystallography of #4V$_L$ tetramer

The #4V$_L$ tetramer was crystallized at 4 °C using the sitting drop vapor diffusion method, with the assistance of crystal plates (CrystalEX Second Generation Corning 3552, Hampton Research, CA, USA) and crystal screening kits (Hampton Research). The protein concentration was fine-tuned to 1.2 mM (in terms of the monomer unit) in 10 mM potassium phosphate buffer, pH 7.0. Droplets were formed by combining 1 μL of the protein solution with 1 μL of the reservoir solution, which were subsequently equilibrated. The optimal reservoir solution was identified as 0.1 M potassium sodium tartrate tetrahydrate. Crystals were obtained in the protein solution after incubation at 4 °C for 15 days.

Diffraction data for #4V$_L$ were obtained at the BL45XU beamline at SPring-8, Japan. Crystals were secured on cryo-loops and flash-frozen at 100 K using a nitrogen cryosystem. Data collection was automated via the ZOO system[39], with automatic data processing handled by the KAMO system[40]. The preliminary structure was derived through the molecular replacement method (PHASER)[41], utilizing the atomic coordinates of the #4V$_L$ structure as predicted by AlphaFold2[32]. The PHENIX.REFINE program[42] was used for structural refinement. Manual corrections were made to the molecular model, and water molecules were identified from the electron density map using the COOT program[43]. A summary of the data collection and refinement statistics is provided in Supplementary Table 3.

## Molecular dynamics simulation

To delve into the structural dynamics of #4V$_L$, a computational study was conducted on two systems: the 3D-DS dimer and the #4V$_L$ tetramer. All-atom MD simulation was performed on the refined X-ray crystal structure of the #4V$_L$ tetramer using the AMBER 20 software package[44]. Missing residues in the X-ray crystal structure of the #4V$_L$ tetramer were supplemented using the MODELLER software[45]. The structure of the 3D-DS dimer was derived from chains A and B of the #4V$_L$ tetramer crystal structure. Amino acid protonation states in these structures were assigned by PROPKA, via the PDB2PQR web server[46]. The dimer and tetramer of #4V$_L$ systems were prepared using the ff19SB force fields and saturated with explicit water molecules employing the TIP3P solvation model. Neutralizing ions were added as Na$^{2+}$ under periodic boundary conditions, utilizing the tLeap program[44]. The dimer system was solvated with 18210 water molecules in a box sized 93.758 × 87.183 × 84.508 Å, while the tetramer system was solvated with 19496 explicit water molecules in a box sized 104.886 × 86.303 × 87.672 Å. For structural minimization, each system was subjected to harmonic potentials and subsequently heated to 310 K for 20 ps under canonical ensemble ($NVT$ - $N$: number of partials in the system, $V$: volume, and $T$: temperature) conditions[47,48]. Each system underwent MD simulations for 400 ns in three replications using AMBER 20[44]. The stability of all systems was assessed by calculating the root mean square deviation (RMSD) using the CPPTRAJ module[49] (Supplementary Fig. 8).

The MD trajectory of the #4V$_L$ tetramer was examined to elucidate molecular interactions of the tetramer during MD simulations, utilizing the RING[50] and PyContact[51] tools. In PyContact, criteria for hydrogen bonding interaction analysis included a ≤ 3.5 Å distance between the hydrogen bond acceptor (HBA) and donor (HBD), as well as an HBD-H⋯HBA interaction angle of ≥ 120°. A total of 3,000 snapshots taken from the final 300 ns-MD trajectory of the #4V$_L$ tetramer were analyzed for the quantity of water in the solvent shell using the 'water-shells' command in the cpptraj module[49]. To assess the structural stability of the #4V$_L$ tetramer in solution, we calculated the free energy difference between the #4V$_L$ tetramer and two 3D-DS dimers. For the proteins, atomic charges and Lennard-Jones parameters derived from the GAFF parameters were applied via the antechamber program[44], and the potential energies were assessed without considering protein-solvent interactions. Instead, the solvation free energy correction was evaluated based on the 3D-RISM theory[52], using 100 snapshots from the final 100 ns-MD trajectory. The mean of the resulting solvation free energy correction values was calculated. In the 3D-RISM calculations, the temperature and density of water were set at 300 K and 1.0 g/cm$^3$, respectively. The TIP3P water model, with modified hydrogen parameters (σ = 0.4 Å and ε = 0.046 kcal mol$^{-1}$), was used. A 3D grid spacing of 0.5 Å was assigned for 256 grid points, as implemented in the RISMiCal software package[53]. Various 3D water distribution functions with $g(r) > 2$, 4, and 8 were visualized using the UCSF Chimera version 1.16 program.

## Thermodynamics and kinetics for #4C214A and #4V$_L$ tetramers

The dissociation equilibrium constants of the #4C214A and #4V$_L$ tetramers to monomers were derived from the respective ratios of monomers to tetramers at varying protein concentrations and temperatures. The standard Gibbs free energy change, $\Delta G°$, was calculated by inserting the dissociation equilibrium constant and temperature

into Eq. (1).

$$\triangle G^\circ = - RT \ln K \tag{1}$$

The standard enthalpy change, $\Delta H^\circ$, and the standard entropy change, $\Delta S^\circ$, were estimated using Eqs. (2) and (3), respectively.

$$\Delta H^\circ(T) = \int_{T_h}^{T} \Delta C^\circ_p(T)\, dT \tag{2}$$

$$\Delta S^\circ(T) = \int_{T_s}^{T} \frac{\Delta C^\circ_p(T)}{T}\, dT \tag{3}$$

Here, $\Delta C^\circ_p(T)$ is considered not to vary with temperature, i.e., $\Delta C^\circ_p (T) = \Delta C^\circ_p$. $T_h$ and $T_S$ represent the temperatures at which $\Delta H^\circ$ and $\Delta S^\circ$ equal zero, respectively. $\Delta C^\circ_p (T)$ was obtained by fitting $\Delta G^\circ(T)$ with Eq. (4) using the least squares method[37,54].

$$\Delta G^\circ(T) = \Delta C^\circ_p(1 + \ln T_s)T - \Delta C^\circ_p T \ln T - \Delta C^\circ_p T_h \tag{4}$$

To obtain the dissociation constants of the #4C214A and #4$V_L$ tetramers to monomers, 300 μM #4C214A and #4$V_L$ solutions were diluted 50 times at temperatures ranging from 4–45 °C using PBS buffer. The tetramer ratios' time courses were monitored by SEC with a Superdex 200 10/300 GL column (Cytiva) on the FPLC system (Bio-logic DuoFlow 10) with a flow rate of 0.5 mL/min, a monitoring UV-vis detector settled at 280 nm, and PBS buffer, at a temperature of 4 °C. The proportion of the tetramer as a function of time was plotted, and the plot was then fitted to an exponential curve using the least squares method.

## Circular dichroism measurements

#4$V_L$ solutions at 10 μM and 100 μM were prepared with 20 mM potassium phosphate buffer, pH 7, and incubated at 20 °C for 24 h to reach equilibrium. The CD spectra of #4$V_L$ solutions were then recorded using a J-820 CD spectropolarimeter (Jasco). The measurements were conducted in 20 mM potassium phosphate buffer, pH 7.0, at 20 °C, with a response of 2 s, spectral bandwidth of 1 nm, and a scan speed of 0.5 nm/min. Quartz cells with path-lengths of 0.1 and 0.01 cm were used for protein concentrations of 10 and 100 μM, respectively. SEC analysis was performed for the 10 μM and 100 μM #4$V_L$ solutions with a Superdex 200 Increase 10/300 GL column on the HPLC system. The HPLC system was used at a flow rate of 0.3 mL/min with 20 mM potassium phosphate buffer, pH 7, at a temperature of 4 °C.

## Reporting summary

Further information on research design is available in the Nature Portfolio Reporting Summary linked to this article.

## Data availability

The data that support this study are available from the corresponding author upon request. The atomic coordinates of the #4$V_L$ tetramer have been deposited to the Protein Data Bank under accession code 8KAD. Plasmids are available from the corresponding author on a reasonable request. Source data are provided with this paper.

## Code availability

Software and codes used to run and analyze MD simulations are listed in Supplementary Table 4 with accession links.

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

## Acknowledgements
This work was partially supported by Grants-in-Aid from JSPS for Scientific Research (B) (No. JP26288080 (S.H.)) and Transformative Research Area "Mesohierarchy" (No. 23H04879 (S.Y.)), and by JST CREST, Japan (No. JP20338388 (S.H.)). We are grateful to the staff at beamline BL45XU, SPring-8, Japan (Proposal No. 2021B2534 and No. 2022A2534) and Dr. Tomoshige Ando, Kyoto Pharmaceutical University, for LC/ESI-QTOF-MS measurements. Numerical calculations were performed using Cygnus at the CCS, University of Tsukuba supported by Multidisciplinary Cooperative Research Program in CCS, University of Tsukuba and a high-performance computing infrastructure project (grant number hp200157 (Y.S.)).

## Author contributions
S.H. conceived the project; T.S. expressed and purified proteins with advises from T.M., N.K., T.U., and E.H.; T.S., T.M., N.K., and S.H. analyzed the experimental data; H.O. performed the X-ray structural analysis. L.D., K.H., R.F., and Y.S. performed theoretical calculations; all authors were involved to discuss the data; T.S., K.H., Y.S., and S.H. wrote the manuscript and all the authors reviewed it.

## Competing interests
The authors declare no competing interests.
