## [Peer Review File · Nature Communications]

Structural and Thermodynamic Insights into Antibody Light Chain Tetramer Formation through 3D Domain SwappingREVIEWER COMMENTS

Reviewer #1 (Remarks to the Author):

The manuscript encoded 23-35268 and entitled:

Structural and Thermodynamic Insights into Antibody Light Chain Tetramer Formation through 3D Domain Swapping

submitted by T. Sakai and coll. is an interesting contribution on the #4 antibody light chain auto-association driven by the 3D domain swapping (DS) mechanism together with hydrophobic interactions. The coupling of these two events was in the past ascribed especially to runaway domain swapping phenomena (McParland L et al 2018, Nat Str Mol Biol 25, 705-14), not for circular oligomers like is in the present case.

Considering the high importance of the monoclonal antibody/ies therapeutical applications, their stability is one of the most important weaknesses limiting their successful use against cancers and viruses.

The group led by prof S. Hirota is worldwide leader in the studies on how 3D-DS governing the auto-association of many cytochromes. This study is instead the first that provides precise structural elements explaining why antibodies are hijacked toward undesired aggregation and functional loss. Moreover, the authors demonstrate that this undesired destiny is ascribable exclusively to the variable region #4VL, not to the constant one, #4CL.

Therefore, the manuscript certainly deserves to be considered for publication in "Nature Communication", pending some amendments and better explanation of some points listed here below:

-The main point may be a consideration for future investigations: since the VL is the responsible, the phenomenon could be difficult to be settled for other antibodies. However, although 3D-DS is independent on the nature of the swapped domain(s), could common features exist for other antibodies to prevent or foresee their self-association?

-Many works have been performed conjugating different VLs with many RNases, such as RNase A, HP-RNase (RNase 1), RNase 5 (angiogenin), or onconase (listed in Gotte & Menegazzi 2019, Front Immun 10, 2626). Could aggregation represent the real limit for the successful use of these VL derivatives, or, alternatively, may this hetero-association prevent VL auto-association? This aspect could be mentioned in the Intro or in the Discussion.

-Another point that could be (shortly) discussed, although not easily solvable: the #4 tetramer is not a crystallographic artifact (chromatography confirm this) and it is formed by the hydrophobic association of two domain-swapped dimers. Since the dimer is not detected in solution, can be now stated that 3D-

DS and hydrophobic interactions occur contemporarily or synergistically? Or, alternatively, can 3D-DS be the event triggering the complete tetramer association through hydrophobic interactions?

By the way, could cystatin-C and cystatins, producing intermediate totally or partially domain-swapped tetramers prior to fibrillation (see, among many others: Sanders A et al 2004, *J Mol Biol* 336, 165-78; Wojciechowska D et al 2022, *Int J Mol Sci* 23, 13441;) comparable and/or helpful? I mention cystatins because the models built for cystatin-C tetramer(s) are similar to the structure tetramer of #4VL solved here (Fig. 2).

Other minor points:

- 1) Abstract: a short sentence is required to better link the first sentence with the second stating that antibodies easily form aggregates.
- 2) Introduction, page 3, 3rd line from the bottom: 3D-DS involves also amyloidogenic proteins (Bennett MJ et al 2006, *Structure* 14, 811-24), this could strengthen the whole phrase. Then 2nd line from the bottom “several types of 3D-DS”? Although an example is reported thereafter, what is exactly meant for “several types”? Runaway DS (T7 endonuclease I, Guo & Eisenberg 2006, *PNAS* 103, 8042-7), propagated DS (cystatin C, Walbhom M et al 2007, *JBC* 282, 18318-26) or stacking of swapped dimers (GB1 mutant, Nagarkar RP et al 2010, *Biopolymers* 94, 141-55)? Other types of combinations of domain swapping-driven interactions are listed in: McPartland L et al (2018, *Nat Str Mol Biol* 25, 705-14).
- 3) Page 4: is there a reference for the “Kabat method”?
- 4) Page 5, 8th line “corresponded well”: “Well” does not sound scientifically good...
- 5) Page 6, Fig. 1: the schematic figures of panels a, e, i should be a little bigger to make the comprehension more immediate. Then, although the result is clear, no explanation is given why with the complete antibody light chain a Superdex 200 Incr 5/150 SEC column has been used, while with #4VL and #4CL a Superdex 200 10/300 (page 7). The reason could be the amount available in the two different cases, if so this should be reported, in the Methods or in the text or in the legend.
- 6) Page 8, 5th line from the bottom: add “free” before variable region.
- 7) Page 10: 6th line, pay attention, Ref. n 28 is the same of n. 21! Then, if VHH and RNase A are mentioned, also β 2-microglobulin (Ref. 29) should be explicitly listed.
- 8) Page 13: all the correct arguments should report that the key-interactions governing the 3D-DS mechanism are the closed interface, present in both the monomer and in the newly forming oligomer, and the open interface, present only in the domain-swapped structure(s) (Bennett M et al 1995, *Protein Sci* 4, 2455-68). In particular, the first two lines refer to the closed interface, while the rest of the page, if I understand well, refers to the open interface.
- 9) Page 18, CDR (and thereafter CDR3) should be explained better.
- 10) Page 20, 6th line from the bottom: “mol-1” is missing in the second deltaS of melittin tetramer.
- 11) Page 22, legend of Fig. 4, panel a: amend ‘Plots’ with “Plot”.
- 12) Methods, page 27 line 1: add “UV-vis detector settled at 280 nm” after monitoring. This should be amended throughout the whole manuscript (Methods, see page 29, 30...). Then, line 5: change ‘but with a lower’ with “at a”. Then, 5th line from the bottom: delete “as” after conditions.
- 13) Page 28, “Expression and purification...” paragraph: anticipate “via mutagenesis” to the first line before “between”. Then, 4th line: the primers should be added in the Supplementary Data. Then, 7th line from the bottom: add “the” before TEV. 5th line from the bottom: delete “treated” and add “then” after was.

- 14) Page 29, first lines: amend with "...anion exchange chromatography performed with a HiTrap Q HP column (Cytiva)...". Then, 2nd line from the bottom: change "by conducting" with "with".
- 15) Page 30, 6th line: has PBS been used to equilibrate the column?. Then, 7th line, "0.30 mL/min". Then, 5th line from the bottom: change 'reconcentrated' with "concentrated again".
- 16) Page 31, 2nd line: change 'these peak...' with "the relative peak areas". Then "at concentrations" not 'in'. 5th line: amend to "SEC-MALS of purified #4VL and #4CL was performed...". Then, last line of the paragraph: "dn/dc" should be better explained.
- 15) Page 36: Circular dichroism: conditions used for near- and far-UV CD should be more explicitly reported.

Reviewer #2 (Remarks to the Author):

The manuscript by Sakai et al presents an interesting study on an antibody light chain (LC) that forms tetramers via VL-domain swapping. The detailed structural and thermodynamic analysis provides mechanistic insights into VL tetramer formation. The authors mention several times the relevance of their findings to the fields of antibody therapeutics and light chain amyloidosis. However, the only LC that they investigated is neither from a therapeutic antibody nor from a patient with light chain amyloidosis or multiple myeloma. Therefore, it is unclear whether the reported domain-swap mechanism is a general feature and whether it has any relevance to antibody therapeutics or diseases. Overall, it is a nice paper and addressing the suggested points will make it even more important.

Specific points:

1. Does the domain-swap stabilize VL and LC against aggregation and amyloid formation?
2. At least one additional VL/LC preferably involved in multiple myeloma or light chain amyloidosis should be included and it would be very interesting to see whether the swap has an impact on the aggregation?
3. Does the VL domain swap also occur in the context of the assembled IgG antibody?
4. Why was the C214A LC mutant used? In natural LC dimers, this cysteine can form an intermolecular disulfide bond that can stabilize a homodimer.

Minor comments:

5. "Molecular weight" is used to describe the mass of the proteins in SEC-MALS and SDS-Page, but if the values are given in kDa, the appropriate term is "molecular mass".
6. Please provide the sequences of all proteins and mutants in the supplementary information.
7. It is widely accepted to write the "L" in "VL" and "CL" in subscript.
8. There is not "X" version of the Astra software (methods part). Please check.

Our Views on the Comments by Reviewer #1:

The manuscript encoded 23-35268 and entitled:

Structural and Thermodynamic Insights into Antibody Light Chain Tetramer Formation through 3D Domain Swapping

submitted by T. Sakai and coll. is an interesting contribution on the #4 antibody light chain auto-association driven by the 3D domain swapping (DS) mechanism together with hydrophobic interactions. The coupling of these two events was in the past ascribed especially to runaway domain swapping phenomena (McParland L et al 2018, Nat Str Mol Biol 25, 705-14), not for circular oligomers like is in the present case.

Considering the high importance of the monoclonal antibody/ies therapeutical applications, their stability is one of the most important weaknesses limiting their successful use against cancers and viruses.

The group led by prof S. Hirota is worldwide leader in the studies on how 3D-DS governing the auto-association of many cytochromes. This study is instead the first that provides precise structural elements explaining why antibodies are hijacked toward undesired aggregation and functional loss. Moreover, the authors demonstrate that this undesired destiny is ascribable exclusively to the variable region #4VL, not to the constant one, #4CL.

Therefore, the manuscript certainly deserves to be considered for publication in “Nature Communication”, pending some amendments and better explanation of some points listed here below:

Answer:

We appreciate your valuable comments and revised the manuscript accordingly.

-The main point may be a consideration for future investigations: since the VL is the responsible, the phenomenon could be difficult to be settled for other antibodies. However, although 3D-DS is independent on the nature of the swapped domain(s), could common features exist for other antibodies to prevent or foresee their self-association?

Answer:

Thank you for the valuable comments. Yes. We believe so. We have been searching for antibody light chains that can undergo 3D-DS. In fact, we have found several antibody light chains that show properties similar to those observed in #4VL. However, we need to obtain the crystal structure to prove 3D-DS, which is difficult for VL because its structure is very flexible. We are trying hard and hope to obtain some more domain-swapped crystal structures of VL in the near future.

-Another point that could be (shortly) discussed, although not easily solvable: the #4 tetramer is not a crystallographic artifact (chromatography confirm this) and it is formed by the hydrophobic association of two domain-swapped dimers. Since the dimer is not detected in solution, can be now stated that 3D-DS and hydrophobic interactions occur contemporarily or synergistically? Or, alternatively, can 3D-DS be the event triggering the complete tetramer association through hydrophobic interactions?

Answer:

The hydrophobic residues that are important for the stabilization of the tetrameric structure are on the surface of the protein. Thus, we speculate that hydrophobic interactions may also exist between monomers. In our preliminary MD calculations of the 3D-DS dimer, the dimer tends to bend, due to the hydrophobic interface. As stated in the manuscript, hydrogen bonds encircled the aromatic cluster (Fig. 3c), presumably augmenting hydrophobic interactions and stabilizing the tetramer structure. The 3D-DS structure makes this possible, which is likely to be important for the stabilization of the tetramer. This is now stated in the revised manuscript. (Page 17 lines 4–3 from bottom)

By the way, could cystatin-C and cystatins, producing intermediate totally or partially domain-swapped tetramers prior to fibrillation (see, among many others: Sanders A et al 2004, J Mol Biol 336, 165-78; Wojciechowska D et al 2022, Int J Mol Sci 23, 13441;) comparable and/or helpful? I mention cystatins because the models built for cystatin-C tetramer(s) are similar to the structure tetramer of #4VL solved here (Fig. 2).

Answer:

Thank you for the comments. Although both #4VL and cystatin-C form tetramers, the interaction between the dimers in each tetramer seem different, resulting in different structures. The 3D-DS dimers of the #4VL tetramer interact symmetrically between each other in the entire region, forming a compact tetrameric structure. On the other hand, according to the cystatin-C tetramer model structure obtained by SAXS measurements, the dimers interact through only one cystatin-C function unit of each dimer, resulting in a little long and narrow structure. However, the relationship of 3D-DS and fibrillation is important. There have been studies suggesting that protein dimerization by 3D-DS tends to accelerate aggregation into amyloid-like fibrils. This is now stated in the introduction section of the revised manuscript. (Page 3, line 2–1 from bottom)

Other minor points:

1) Abstract: a short sentence is required to better link the first sentence with the second stating that antibodies easily form aggregates.

Answer:

We have modified the link between the first and second sentences.

2) Introduction, page 3, 3rd line from the bottom: 3D-DS involves also amyloidogenic proteins (Bennett MJ et al 2006, Structure 14, 811-24), this could strengthen the whole phrase.

Then 2nd line from the bottom “several types of 3D-DS”? Although an example is reported thereafter, what is exactly meant for “several types”? Runaway DS (T7 endonuclease I, Guo & Eisenberg 2006, PNAS 103, 8042-7), propagated DS (cystatin C, Walbhom M et al 2007, JBC 282, 18318-26) or stacking of swapped dimers (GB1 mutant, Nagarkar RP et al 2010, Biopolymers 94, 141-55)? Other types of combinations of domain swapping-driven interactions are listed in: McPartland L et al (2018, Nat Str Mol Biol 25, 705-14).

Answer:

“types” is corrected to “structures” to be more accurate. The sentence now reads as “Several 3D-DS structures have also been reported for certain antibodies”.

3) Page 4: is there a reference for the “Kabat method”?

Answer:

We have added a reference for the Kabat method (E. A. Kabat, *et al.*, Sequences of proteins of immunological interest, NIH publication, U.S. Dept. of Health and Human Services, Public Health Service, National Institutes of Health, 1991)

4) Page 5, 8th line “corresponded well”: “Well” does not sound scientifically good...

Answer:

The phrase is modified to “were in good agreement to”.

5) Page 6, Fig. 1: the schematic figures of panels a, e, i should be a little bigger to make the

comprehension more immediate. Then, although the result is clear, no explanation is given why with the complete antibody light chain a Superdex 200 Incr 5/150 SEC column has been used, while with #4VL and #4CL a Superdex 200 10/300 (page 7). The reason could be the amount available in the two different cases, if so this should be reported, in the Methods or in the text or in the legend.

Answer:

The schematic figures of panels a, e, i are enlarged.

We have performed the experiments of the complete antibody light chain with a Superdex 200 10/300 column for panels b and d to make the column conditions the same as the other experiments.

6) Page 8, 5th line from the bottom: add “free” before variable region.

Answer:

We added “free” as suggested.

7) Page 10: 6th line, pay attention, Ref. n 28 is the same of n. 21! Then, if VHH and RNase A are mentioned, also β 2-microglobulin (Ref. 29) should be explicitly listed.

Answer:

We have added β 2-microglobulin and corrected the references.

8) Page 13: all the correct arguments should report that the key-interactions governing the 3D-DS mechanism are the closed interface, present in both the monomer and in the newly forming oligomer, and the open interface, present only in the domain-swapped structure(s) (Bennett M et al 1995, Protein Sci 4, 2455-68). In particular, the first two lines refer to the closed interface, while the rest of the page, if I understand well, refers to the open interface.

Answer:

Since the swapping region is a β -strand in a β protein, most interactions occur at the closed interface and interactions at the open interface are rare. There is only one new interaction in the present case (HB 7). However, this interaction is a little different from the definition of the open interface interaction, since HB 7 is formed between protomers; not within the same protomer.

9) Page 18, CDR (and thereafter CDR3) should be explained better.

Answer:

“CDR” is corrected to “CDR3”. CDR3 is explained in page 10, and a brief explanation on the hydrophilic property of CDR3 is added to the revised manuscript. (Page 18, lines 5–3 from bottom).

10) Page 20, 6th line from the bottom: “mol⁻¹” is missing in the second deltaS of melittin tetramer.

Answer:

“mol⁻¹” is added to the revised manuscript.

11) Page 22, legend of Fig. 4, panel a: amend ‘Plots’ with “Plot”.

Answer:

The figure legend is corrected.

12) Methods, page 27 line 1: add “UV-vis detector settled at 280 nm” after monitoring. This should be amended throughout the whole manuscript (Methods, see page 29, 30...). Then, line 5: change ‘but with a lower’ with “at a”. Then, 5th line from the bottom: delete “as” after conditions.

Answer:

The text is modified throughout the whole manuscript as suggested.

13) Page 28, “Expression and purification...” paragraph: anticipate “via mutagenesis” to the first line before “between”. Then, 4th line: the primers should be added in the Supplementary Data. Then, 7th line from the bottom: add “the” before TEV. 5th line from the bottom: delete “treated” and add “then” after was.

Answer:

The manuscript is corrected.

The information of primers is added to the Supplementary Information.

14) Page 29, first lines: amend with "...anion exchange chromatography performed with a HiTrap Q HP column (Cytiva)...". Then, 2nd line from the bottom: change "by conducting" with "with".

Answer:

The manuscript is corrected.

15) Page 30, 6th line: has PBS been used to equilibrate the column?. Then, 7th line, "0.30 mL/min". Then, 5th line from the bottom: change 'reconcentrated' with "concentrated again".

Answer:

PBS has been used to equilibrate the column. We performed additional experiments using the Superdex 200 Increase 10/300 GL column with a flow rate of 0.3 mL/min, instead of 0.25 mL/min. Thus, we left the expression of the flow rate as "0.3 mL/min".

16) Page 31, 2nd line: change 'these peak...' with "the relative peak areas". Then "at concentrations" not 'in'. 5th line: amend to "SEC-MALS of purified #4VL and #4CL was performed...". Then, last line of the paragraph: "dn/dc" should be better explained.

Answer:

The manuscript is corrected. "dn/dc" is modified to "refractive index increment (dn/dc)" in the revised manuscript.

15) Page 36: Circular dichroism: conditions used for near- and far-UV CD should be more explicitly reported.

Answer:

We added the conditions of the circular dichroism to the Methods section.

Our Views on the Comments by Reviewer 2:

The manuscript by Sakai et al presents an interesting study on an antibody light chain (LC) that forms tetramers via VL-domain swapping. The detailed structural and thermodynamic analysis provides mechanistic insights into VL tetramer formation. The authors mention several times the relevance of their findings to the fields of antibody therapeutics and light chain amyloidosis. However, the only LC that they investigated is neither from a therapeutic antibody nor from a patient with light chain amyloidosis or multiple myeloma. Therefore, it is unclear whether the reported domain-swap mechanism is a general feature and whether it has any relevance to antibody therapeutics or diseases. Overall, it is a nice paper and addressing the suggested points will make it even more important.

Answer:

We appreciate the reviewer's valuable comments. We believe that 3D-DS occurs at least in several VL and LC. In fact, we have found that other LCs can exhibit the 3D-DS character similar to #4 LC, although we have not yet solved the crystal structures.

【Specific points】

1. Does the domain-swap stabilize VL and LC against aggregation and amyloid formation?

Answer:

Although there is no promising data on the relationship between 3D-DS and aggregation in antibodies, #4VL undergoes 3D-DS and its structure is more stable in the tetrameric form than in the monomeric form, according to the CD spectra at different concentrations (Fig. 4i). Thus, the occurrence of aggregation and/or amyloidosis may decrease by 3D-DS. On the other hand, there have been several reports discussing the relationship between 3D-DS and amyloid fibril formation, where the protein dimerization by 3D-DS tends to accelerate aggregation into amyloid-like fibrils (J. M. Louis, *et al.*, *J. Mol. Biol.* **348**, 687-689 (2005); M. Wahlbom, *et al.*, *J. Biol. Chem.* **282**, 18318-18326 (2007); E. Zerovnik *et al.*, *FEBS J.* **278**, 2263–2282 (2011)). This is added to the introduction of the revised manuscript. (Page 3, lines 2–1 from bottom)

2. At least one additional VL/LC preferably involved in multiple myeloma or light chain amyloidosis should be included and it would be very interesting to see whether the swap has an impact on the aggregation?

Answer:

Thank you for your comments. We understand your opinions. However, most of the 3D-DS structures are not stable in proteins, especially in VL and LC, since they are flexible. Although we have been investigating many VL and LC, #4VL is the rare case that 3D-DS can be identified. As written in the manuscript, the 3D-DS structure of #4VL was obtained, owing to the stabilization of the 3D-DS dimer through tetramer formation driven by hydrophobic interactions between two 3D-DS dimers.

Regarding the light chains extracted from multiple myeloma patients, the idea is quite interesting, but it requires assistance from medical doctors. As previously mentioned, 3D-DS structures are not stable. Even if we can obtain samples, it will be a hard task using X-ray crystallography to identify the light chain that can undergo 3D-DS.

3. Does the VL domain swap also occur in the context of the assembled IgG antibody?

Answer:

This is an important point. As described in the Introduction section, human antibody 2G12 undergoes 3D-DS by exchanging the heavy chain variable region (Ref. 19). VHH dimerizes by performing 3D-DS for the C-terminal β -strand (Ref. 20). In addition, another report (Y. Luo, *et al.*, *MAbs*, **9**, 916–926 (2017): Evidence for intermolecular domain exchange in the Fab domains of dimer and oligomers of an IgG1 monoclonal antibody) suggests 3D-DS of IgG/Fab by exchanging the whole VL between two Fab regions. This is added to the discussion of the revised manuscript (Page 4, lines 3–4). Our present work reveals another type of 3D-DS with a detailed 3D structure, where a single β -strand is exchanged between two VL proteins.

As 3D-DS can occur in various manners in antibodies, 3D-DS may accelerate associate. This is added to the discussion of the revised manuscript. (Page 24, line 2 from bottom – Page 25, line 3)

4. Why was the C214A LC mutant used? In natural LC dimers, this cysteine can form an intermolecular disulfide bond that can stabilize a homodimer.

Answer:

We have two reasons for using the C214A LC mutant in this study. The first reason is that we need to prepare VL over 10 mg/mL in concentration for crystallization. #4VL is produced by the cleavage of the C214A LC mutant, where a large quantity can be prepared with high expression and recovery. The other reason is to remove the effect of structural mixtures of LC

generated in the system. If the 3D-DS dimer is present with the dimer produced by the disulfide bond of two cysteines residues of different LC molecules, accurate analysis and interpretation will be difficult. We have added an explanation of using the C214A mutant to the revised manuscript. (Page 4, lines 5–3 from bottom)

However, the presence of 3D-DS together with disulfide bond linkages of two cysteine residues of different antibody light chains may promote aggregation. This is now mentioned in the revised manuscript. (Page 24, line 2 from bottom – Page 25, line 3)

【Minor comments】

5. “Molecular weight” is used to describe the mass of the proteins in SEC-MALS and SDS-Page, but if the values are given in kDa, the appropriate term is “molecular mass”.

Answer:

“molecular weight” is correct to “molecular mass” throughout the manuscript.

6. Please provide the sequences of all proteins and mutants in the supplementary information.

Answer:

The sequences of the proteins and primers are provided in the supplement information.

7. It is widely accepted to write the "L" in “VL” and “CL” in subscript.

Answer:

L of VL and CL are changed to subscripts throughout the manuscript.

8. There is not “X” version of the Astra software (methods part). Please check.

Answer:

“X” is removed from the text.